# Evaluating the Influence of Data Entropy in the Use of a Smart Equipment for Traffic Management at Border Check Point

Florin Rusca [1,2,*], Aura Rusca [1,2], Eugen Rosca [1,2], Catalin Coman [1,2], Stefan Burciu [1] and Cristina Oprea [1]

1  Faculty of Transport, University Politehnica of Bucharest, Spl Independentei, No 313, 060042 Bucharest, Romania
2  Inteligent Convergent Solutions, Washington No 39, 011793 Bucharest, Romania
*  Correspondence: florin.rusca@upb.ro

**Abstract:** The transit through a Border Check Point of cargo vehicles supposes, in the case of the Romanian highway network, the carrying out of a process of weighing and verifying of transport licenses. The limited number of weighing equipment and the long duration of these processes cause large queues and long waiting times. A solution for these problems is to use smart equipment to identify the cargo vehicles and to separate the vehicles that require weighing from exempted ones. The separation process is made using external input data. The quality of received data can generate some dysfunctionality in the separation process. The discrete simulation model can be used to evaluate the influence of the uncertainty over the system serving parameters. A study case is developed for a real situation using real data collected from a Romanian Highway Traffic Control Center (HTMC). The results are used in the implementation of the new smart equipment in a Romanian Border Check Point.

**Keywords:** uncertainty; traffic data; smart equipment; discrete simulation model

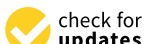



## 1. Introduction

Border Check Points (BCPs) are important for national road networks because they ensure the connection with other national road networks. In the case of the Romanian Highway Network, some specific processes are made for monitoring the cargo entry and exit flows in/from the country. A scaling process is made using dedicated equipment for all cargo vehicles. According to national legislation, it is mandatory to weigh all vehicles from outside the European Union or the ones that exceed the maximum allowed weight. The lack of information regarding the country of origin and their condition obliges the national authority to carry out the weighing process for all cargo vehicles. The reduced number of weighing equipment and, respectively, the long duration of the process, leads to long queues of cargo vehicles. Thus, the waiting time in BCPs can increase to high values. For the case of BCP Nadlac, analyzed in this paper, the waiting queue stretches for approximately 10 km. Cargo vehicles are obliged to wait up to 48 h.

To reduce this negative impact of the weighing process, the introduction of specially designed smart equipment can be a solution. The authors of this paper were asked to design a system based on smart equipment that would allow the separation of flows of cargo vehicles according to a set of rules established by the national authority. The input data in the logical separation process are obtained from the system's external sensors that can record vehicle numbers (License Plate Recognition camera-LPR). A second flow of input data is obtained from the Highway Traffic Management Center (HTMC). The last flow will contain information regarding the plate number, the number of axles, the weight on the axle and the total weight for vehicles.

The analysis of the external data flow provided by HTMC showed the existence of some uncertainty regarding the method by which to identify cargo vehicles. The system

designed by the authors of the paper, Smart Hub, must have the ability to select the vehicles for which the weighing process is mandatory. For these select vehicles, the external data flow can provide uncertain information. The entropy of the external data received by the system can be evaluated based on the amount of information present in the variable of input data flow. Mathematically, the entropy of a random variable is the average level of "information", "surprise", or "uncertainty" inherent in the variable's possible outcomes. If the "uncertainty" raises, the values of the entropy can also raise. For a system developed for the control of traffic flows, it is important to evaluate the entropy and the impact of uncertainty. In our research, the uncertainty is generated by the altered data about cargo vehicles. HTMC sensors are used to collect data through a dynamic scaling of cargo vehicles in two locations from the highway near BCP. The weather conditions, the asphalt quality and the malfunctions of the sensors can alter the quality of the collected data.

The testing process of the logical solution is carried out using a discrete simulation model. With the help of this model, the queuing system parameters can be determined. The structure of the model is made in relation to the real situation of a BCP located between Romanian and Hungarian road networks. A set of simulation scenarios were introduced in the paper. Based on the results obtained, an analysis of the influence of the introduction of smart equipment is made for the control of goods traffic in BCPs, respective of the uncertainty of the input data in the smart equipment separation algorithm.

## 2. Literature Review

The modeling of highway traffic is one of the most important activities for researchers and specialists in the field of transport. This activity helps to understand the process, the importance of data used in the model and the influence over the results [1,2]. The impact assessment for the influence of endogenous and exogenous random events with consequences for users provides the transport infrastructure administration the opportunity to implement a new system based on smart equipment for traffic monitoring and control [3]. The smart concept is reflected by the new capabilities of this equipment that are of interest to experts. The highway infrastructure can be integrated into a complex evolutionary and adaptive system to new situations generated by traffic entities, environmental factors, and legislative decisions [4].

The research was guided to a traffic management of cargo flows in the BCP area. The data regarding cargo vehicles is obtained using dedicated sensors and is used to develop the new traffic management system and logical model associated [5–7]. In some cases, the required data is obtained from external sources such as crowdsourced data to create freight corridors using the connection between vehicles and smart equipment installed on transport infrastructure [8]. In other cases, the development of IoT components dedicated to the automotive industry allows the creation of system architecture that encompasses the integration of an IoT sensor. The traffic management equipment developed using this technology allows the implementation of reliable algorithm for data acquisition and processing [9–11].

The second part of our research is the development of the simulation model for the main activities made presently in a real BCP area. The model is upgraded for a proposed topology and new process made after the implementation of a dedicated smart equipment. Taking into consideration the cargo and other type of vehicle flows interaction, the non-uniform arrivals of vehicles and discrete event simulation stands as a feasible technique for investigating the serving stations capacity during the actual planning stage of the BCP area or for the proposed topology [12,13]. The software tools such as Aimsun, Vissim or Arena Rockwell allow the development of discrete simulation. The last of them allows a logical structure which can be easier to follow and control in research [14–16].

The last part of the research and the main objective of the paper is the assessment of the influence of data uncertainty over the service parameters of the system designed and evaluated by simulation. The incorrect data obtained from sensors that are used to measure the vehicles weight or to recognize the plate number must be identified and corrected

or entered in the modeling as a percentage of incorrect data [17,18]. In this approach the uncertainty is accepted, and the simulation model is used to evaluate the impact of it.

### 3. The Architecture of Smart Equipment and Logical Process

The BCP Nadlac is located between Romanian and Hungarian Highway Networks. The position is in the western part of Romania and is one of the most important connections between the national road network and the EU core transport network (Figure 1). The cargo vehicles are scaled in the actual configuration using two dedicated pieces of equipment. The scaling process includes a border verification for the infrastructure usage fee. The owner of the Romanian Highway Network enforces internal regulations for weighing the cargo vehicles with values over legal limits (e.g., weight or length). The lack of information about cargo vehicles obliges the infrastructure administrator to carry out the weighing process for the entire flow of cargo vehicles. New smart equipment developed for this BCP is proposed to be used in sorting cargo vehicles in two flows. In the first one, the cargo vehicles from the European Union with the weight and length under the legal limit are included. In the second one, the vehicles from countries outside the European Union or with the weight or length over the legal limit or with attributes erroneously collected will be included.

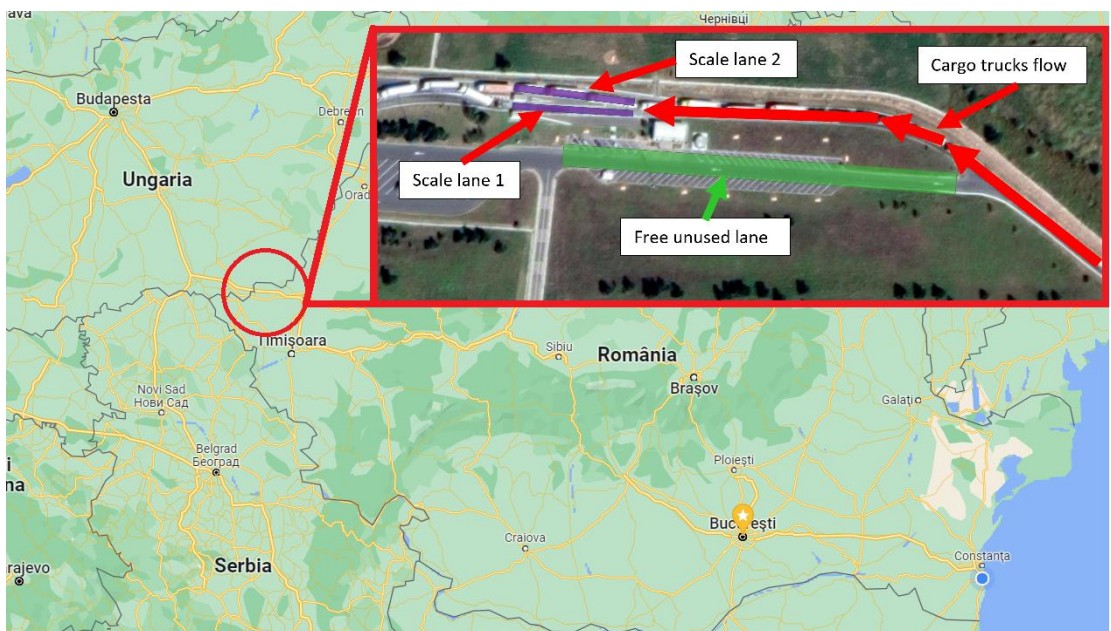

**Figure 1.** The position and the actual configuration of cargo vehicle flows in BCP Nadlac.

The smart equipment is developed by the authors of the paper, members of a research project using funds received from the European Commission. Using a dedicated name, Smart Hub, the common structure of smart equipment contains a set of modules for data analyses, sensors control, sensors data acquisitions, external data transfer and other modules. This structure is depicted in Figure 2. Blue arrows indicate the communications between the central module IoT Hub and all other modules. Green arrows represent the data flows from sensors and from external sources, in our case from HTMC. The connections between modules are ensured using a dedicated module IoT Hub. The sensors proposed to be used in the recording process of cargo vehicle's identity are:

- two License Plate Recognition (LPR) VIDAR cameras located at the entrance in the service area used to identify the plate number of cargo trucks. The technology is provided by ARH commercial company. It is recommended to have integrated illumination, advanced brightness control, and sensor(s) capable of up to 120 FPS. They will be placed in front of and behind the vehicles to reduce the risk of their identification

not being carried out within normal parameters. The camera will have a 20 m range, the vehicle speed range can be between 0 km/h and 300 km/h, and the maximum road width covered must be over 6 m. The sensors resolution is 1440 × 1080. The plate number recognition algorithm is provided by the camera manufacturer. The output data used in the model will be the identification number and time stamp for every vehicle;

- two LPR cameras will be placed on the direct lane to the BCP for a secondary verification and to force the cargo vehicles drivers to use the correct path;
- an inductive loop to verify if the direct lane to BCP is free.

The traffic control equipment proposed to be used by the Smart Hub in the separation process are:

- a Variable-Message Sign (VMS) located in the cargo trucks separation area on two paths: the first path for vehicles from European Union (EU) with vehicle attributes (weight, length, number of car axles) inside the legal limits with a direct lane to BCP; the second path for vehicles with attributes outside the legal limits or from outside countries of the EU or with data not correctly identified (the subject of our research);
- a VMS located on a direct lane to BCP to be used in case of cargo trucks identified on an incorrect path (as a mistake or an attempt to avoid the Scaling Process);
- a traffic light located before the separation point to stop the cargo vehicles for an improved license plate recognition process;
- a terminal for the traffic operator to send alerts about using an incorrect path by cargo vehicles.

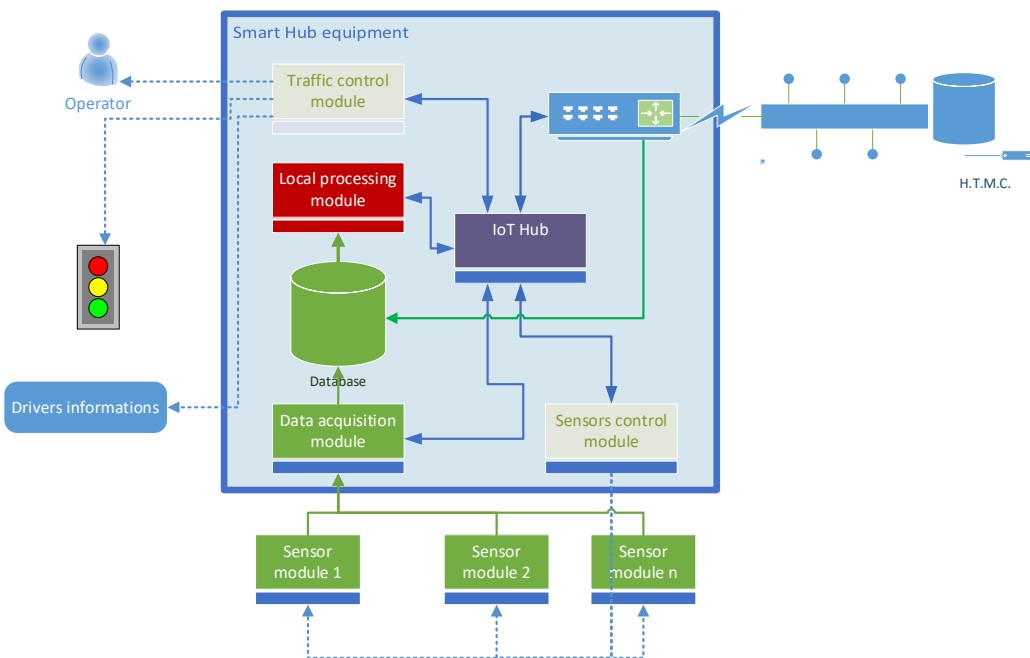

**Figure 2.** The architecture of the proposed smart equipment.

The logical model for the separation process is depicted in Figure 3. The external process to collect information about the license plate of cargo vehicles in correlation with their traffic attributes such as length, weight and number of axles is represented separately on the right of the figure. The data obtained from HTMC are recorded in the Smart Hub database and are classified in four classes:

- vehicle from EU with attribute values inside the legal limits,
- vehicles from EU with attribute values outside the legal limits,
- vehicles outside EU,
- vehicles not correctly identified.

To obtain this data, the Highway Administration uses the dynamic scaling in two locations from the highway before BCP. Two Weigh In Motion (WIM) systems—automated weight enforcement from Kistler—are installed. It is not part of our research, but to improve the quality of data the Highway Administration, we changed the sensors layout diagram from 2 sensors on lane to 5 sensors on lane. The failure of one of the sensors, the modification of the structural parameters of the detection area (rain, snow, deformations of the running surface) can lead to the provision of erroneous data that can be corrected by returning to normal operating parameters. The export data contain information about plate number, weight, number of axles and time stamp.

The data (license plate number) obtained from LPR cameras are introduced in the same database and compared with data (license plate number) obtained from an external source. If the vehicle license plate number is identified in a database, one of the four vehicle classes is assigned to it. If the vehicle is from EU and the traffic attribute values are inside the legal limits, it is guided directly to the BCP. For all other classes, the vehicle will be weighed. In the case that the license plate number is not found in the database, the vehicle will be weighed.

For the vehicles guided on a direct path, a second verification of the license plate number is made to avoid the case where a vehicle does not respect the allocated path. If a non-compliance with the rules is found, an alert will be sent to the operators.

For the actual situation, the scaling process for all cargo vehicles leads to a large queue with waiting time over 48 h in peak traffic periods. Splitting the vehicle flow onto two paths has a positive impact. Considering the scaling process and documents verifications as processes in the queuing system, the increase of the number of service stations leads to the reduction of waiting times. The proposed configuration is presented in Figure 4.

In blue we see the input flow of cargo trucks from the highway, after the selection process. In red, we see the cargo truck flow going to the weighing equipment. In green, we see the cargo truck flow on a direct lane to BCP. Separating these flows, in orange we see the flow of cargo trucks identified on an incorrect path and forced to make a detour to return to the flows selection point. However, the action of separating the flow of vehicles based on some rules may cause certain blockages in the system's activity. If the number of cargo vehicles guided on a direct path to BCP is too big, the scaling process is blocked by the queue made by these ones. In the opposite situation, the large number of cargo vehicles guided to the scaling equipment can block all vehicles, even the ones guided on direct path to BCP. The limited slot for a direct path to BCP is implemented in practice using a space which is not currently in use. This access road will be arranged by the highway administration to allow access according to the rules presented in this paper. The views from the site are present in Figure 5. In the first image, the actual utilization of the free lane is presented. In the initial project, there was supposed to be a service area here. The formation of a queue of cargo vehicles on the highway and on the access connection to the parking lot blocks the access of cars in this area. In images (b) and (c) we see the access to the scaling area and the actual equipment used. For the research, the most important factor is the separation area presented in the last image where the equipment for traffic control and monitoring will be installed.

Another aspect with impact on the system's activity is the uncertainty that may appear in the data obtained from the external source. In Figure 3, the last class used to analyze the data obtained from HTMC is "vehicles not correctly identified". The cause of this type of uncertainty can be the weather conditions (rain, snow, night hours) or the quality of sensors, located on the highway, used by HTMC to collect data on cargo vehicles. From the data collected by the authors, a variation of uncertainty results which can vary from 5% up to 40%. A prediction of the uncertainty cannot be determined, but the activity of Smart Hub equipment can be tested using a discrete simulation model. For different structures of cargo vehicle flows, a set of scenarios can be built to evaluate the influence of uncertainty over the system measures of performance (waiting time, usage ratio of serving stations, length of the queues).

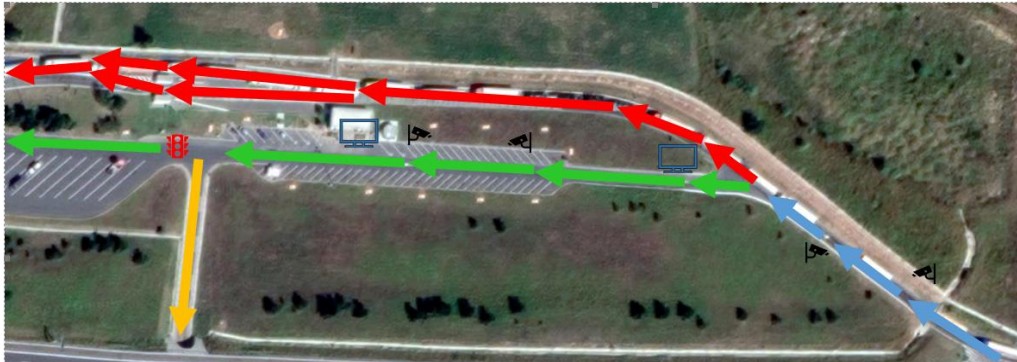

**Figure 3.** The logical model for checking vehicle attributes.

**Figure 4.** The cargo vehicle flow monitored and controlled with Smart Hub equipment.

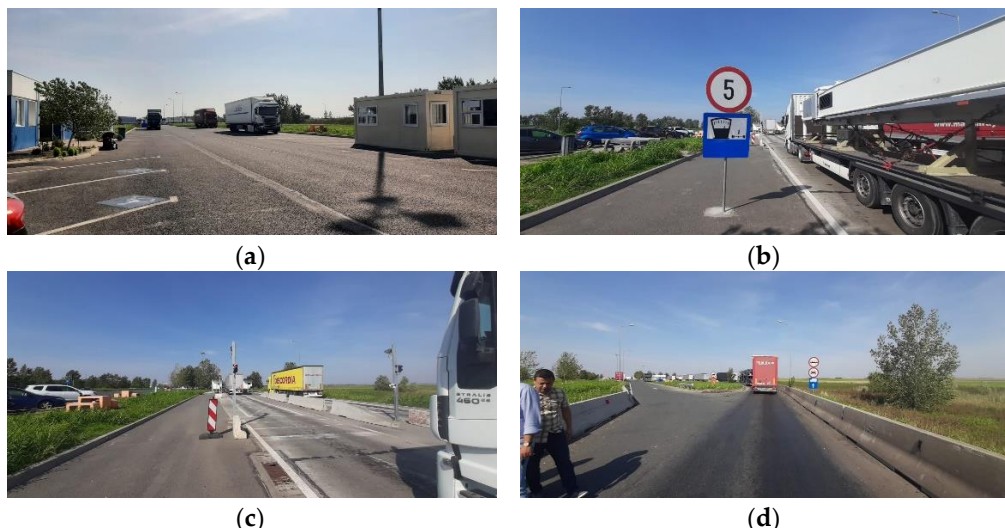

**Figure 5.** The important area from the BCP. (**a**) The unused lane proposed as a new path; (**b**) The scaling area; (**c**) The actual equipment for traffic control; (**d**) The proposed flows separation area.

## 4. The Discrete Simulation Model

The assessment of the activity in a real BCP can be made considering the scaling process, the weighting equipment and the human control team as parts of the queuing system. The cargo vehicles flow is modeled using a theoretical function in accordance with empirical values obtained from HTMC. To calculate the measures of performance of the queuing system, a discrete simulation model using the dedicated software Arena Rockwell was developed. The input data in the simulation model are generated for a set of scenarios to reflect the real situation. The main steps in the developing process of the simulation model and the obtained results are depicted in Figure 6. The calibration process is an important step in ensuring a high degree of confidence in the results obtained through simulation. The parameter used in calibration is the length of the waiting queue for a real situation. Minor changes at the serving time distribution for the scaling process was made to obtain a scaling process queue length with the same values as in reality.

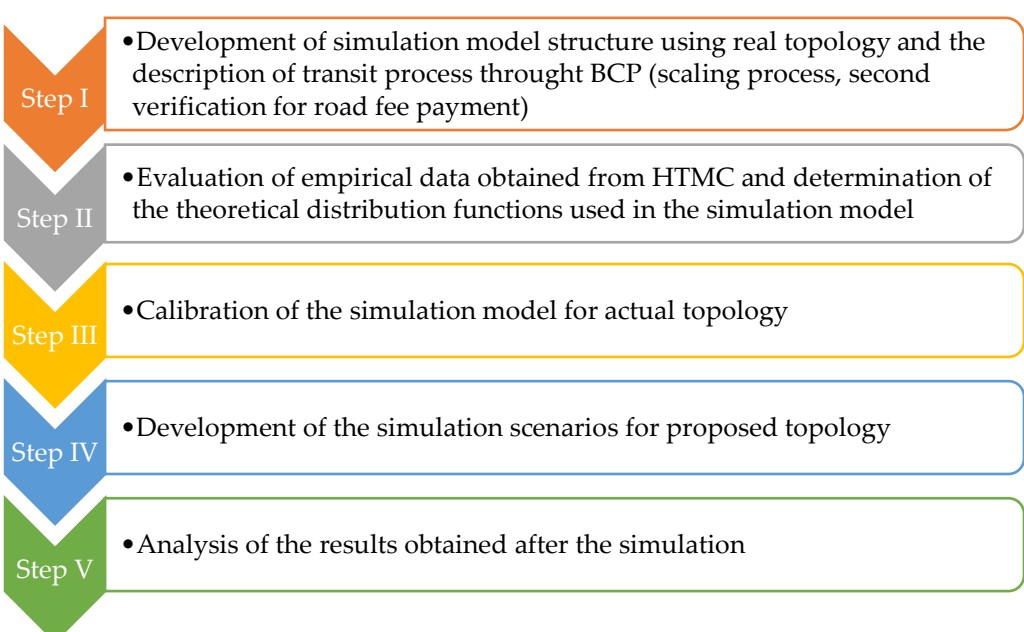

**Figure 6.** The simulation model development steps.

The discrete simulation model is developed using a dedicated software tool known as Arena Rockwell. The software uses the discrete simulation event method to reflect certain real-life processes such as the manufacturing process or activity in transport terminals. Modelling the system as a queue system, the discrete simulation is appropriate. The processes are introduced in the simulation using logical blocks. These are connected in the order of the processes taking place at the control point. The input flow and the main process are represented using logical blocks and are set using real data obtained from HTMC or measured by the authors of this paper. The replication of the model can be made following the model in which the graphic form of the logical blocks is specific to the ARENA software tool. Also, in order to replicate the simulation model, its parameters need to be determined by measurements for each individual case. The simulation model for an actual situation is depicted in Figure 7.

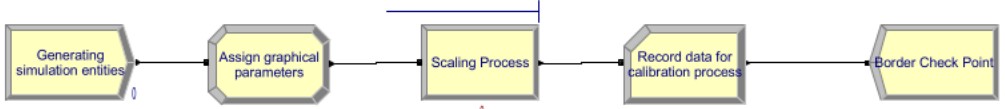

**Figure 7.** The simulation model for an actual situation.

The simulation model for the proposed topology is depicted in Figure 8.

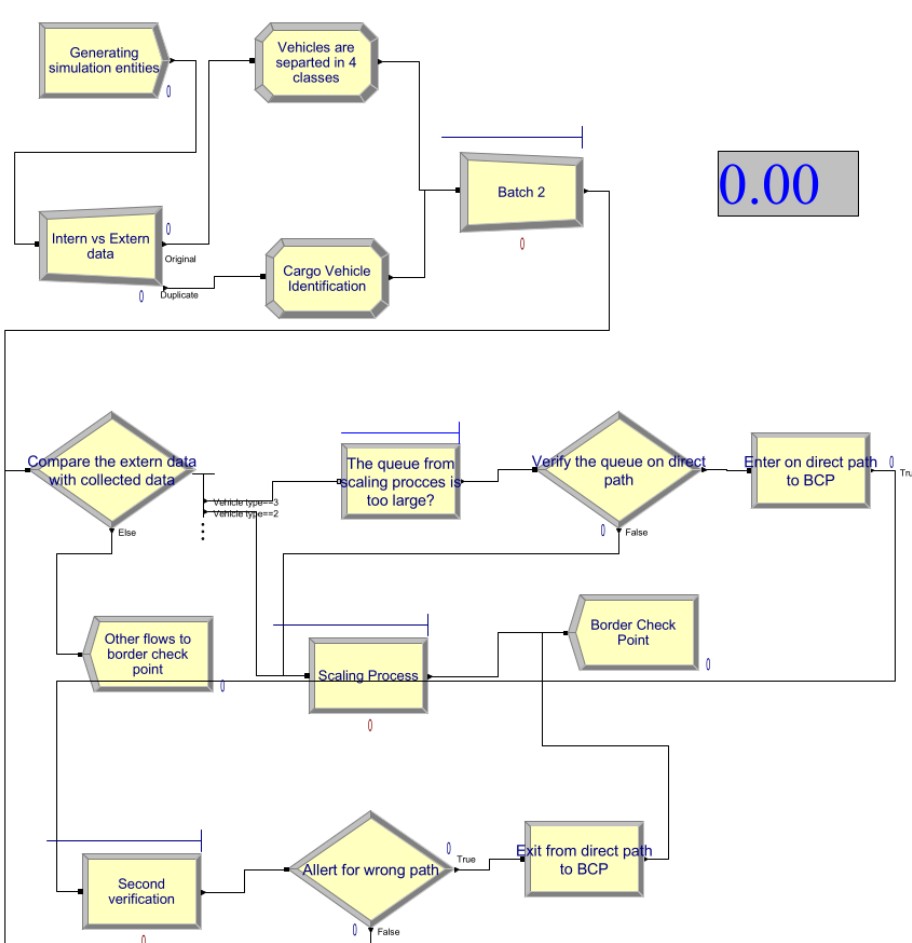

**Figure 8.** The simulation model for the case with Smart Hub equipment implemented.

For every generated simulation entity, one of the four previously established classes is set. The allocation is uniformly made using scenarios parameters. A decision block is used to set the path for cargo vehicle: direct to BCP or over scaling equipment. A hold block is

used to verify if the queue of the guided cargo vehicles over scaling equipment is too large and blocks the direct path to BCP.

For a direct path to BCP, where a second verification of documents, such as a fee for using the Romanian road network, is made by an operator from the Highway Administration, a limited slot for cargo vehicles is set. If the number of cargo vehicles is equal with this number, then no more vehicles are allowed. In this case, the vehicles are routed to scaling equipment unless, according with the previously established conditions, it is not necessary.

The collected data from HTMC are processed so that the theoretical distribution can be determined. The variation in the number of vehicles arriving from the highway in intervals of 10 min is represented in Figure 9. The collection period was from Monday to Friday from the first week of July. On Saturday and Sunday, the activity in BCP is stopped according to regulations of the neighboring country, Hungary.

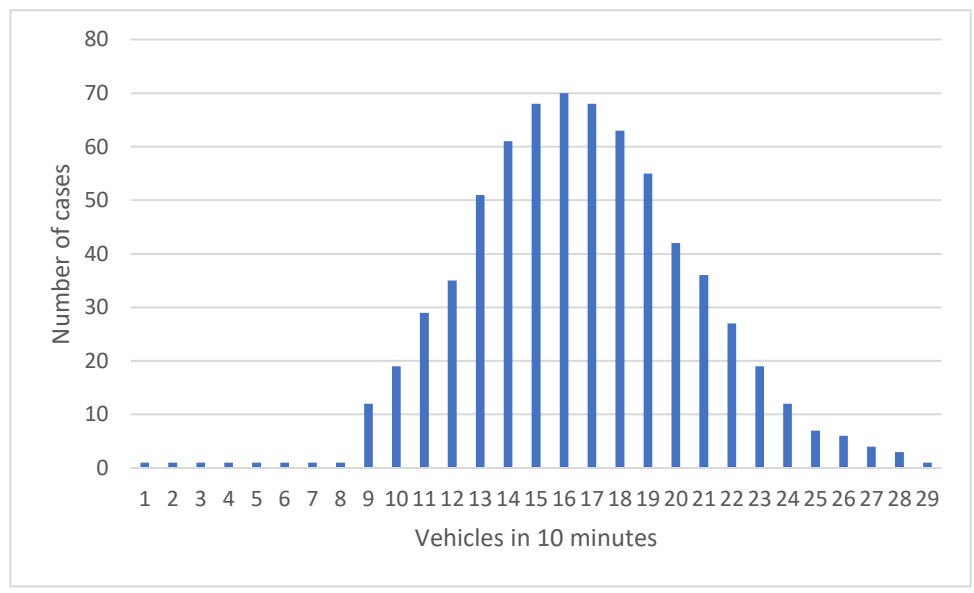

**Figure 9.** The empirical collected data.

The methods used for the modelling of the served traffic units depends on the characteristics of the cargo vehicles flows and of the serving stations. Synthetically, the Kendall–Lee classification [19] use the symbols A/B/n: (m/D) facilitating the models identification. The meaning of that codification is: A is the distribution of the inter-arrival times of entering flow; B—the distribution of the serving times of the traffic units in a serving station/resource; n— the number of serving stations which are working simultaneously; m—the maximum number of places/spots in a serving system for waiting units (maximum queue length); D—the serving discipline or rule. We consider λ the average value of the distribution corresponding to the flow of arrivals and μ the average value for the serving time over weighting equipment. If λ << nμ, then the waiting time before the control point is almost zero; otherwise, in case that λ > nμ, the waiting times before weighting process are significant and queues appear. Because the cargo vehicles arriving are occurring in a fixed interval of time and have a known constant mean rate, the empirical distribution can be approximated with a Poisson distribution. The variance of the distribution is equal with λ (16.7 vehicles in 10 min).

The concordance testing using $X^2$ test are made for the interval between the events time of cargo vehicles arriving. The parameter estimated in the concordance test is the mean of the distribution (λ). The function has a distribution characterized by the number of degrees of freedom f:

$$F = m - k - 1, \tag{1}$$

where m is the number of observed empirical frequencies and k is the number of parameters of the estimated theoretical distribution (k = 1).

The value of $X_c^2$ is calculated using the equation:

$$X_c^2 = \sum_i \frac{(n_i - n_i')^2}{n_i'},\tag{2}$$

where $n_i$ is the empirical frequencies of cargo vehicles arrival and $n_i'$ is the theoretical frequencies of cargo vehicles arrival. The last of them are calculated using the equation:

$$n_i' = N * P(i),\tag{3}$$

where $N$ is the size of the analyzed sample, $P(i)$ is the probability of the arrival of $i$ cargo vehicles in the time unit calculated using the equation:

$$P(i) = \frac{\lambda^i}{i!} e^{-\lambda},\tag{4}$$

Because the calculated value of $X_c^2$ is equal with 9.034 < 12.46, the value of $X_{0,\,f,c}^2$ the value of quantile of function $X^2$ for 28 number of degrees of freedom f and a 95% confidence level for the empiric data obtained from HTMC, a Poisson theoretic distribution can be associated. Due to this association, it can be considered that for the arrivals between freight vehicles, a theoretical exponential distribution can be associated with λ = 36 s.

For the case of a queue system in a stationary regime with multiline serving in parallel stations and distribution of the arrivals in a time interval is approximated using the Poisson distribution, according to the Little equations [20], the average waiting time is:

$$t_a = \frac{P(k = n)n\mu}{(n\mu - \lambda)^2}, \text{ if } \frac{\lambda}{n\mu} < 1,\tag{5}$$

where $P(k = n)$ is the probability of the k cargo vehicles inside the system being equal to the total number of serving stations (in our case we have two scaling equipment) and can be calculated with the equation:

$$P(k = 2) = \frac{\rho^2}{2!} P(0),\tag{6}$$

where $\rho = \frac{\lambda}{\mu}$ and $P(0)$ is the probability that both scaling equipment are free (available) and is calculated with the equation:

$$P(0) = \frac{1}{\sum_{k=0}^{(2-1)} \frac{\rho^k}{k!} + \frac{\rho^2}{(2-1)!(2-\rho)}}\tag{7}$$

Using the previous equations for a serving time constant equal with 75 s, for every cargo vehicle, the average waiting time calculated with equation 6 is equal with 0.42 h. In reality, the serving time is not constant. According to our survey, it can vary around the average value of 75 s. The use of the normal distribution is more suitable. In this case, the mean waiting time can have a larger value which can be determined with the help of the simulation model and then used by comparison with the waiting times collected from drivers in its calibration.

The influence of the uncertainty over the system serving parameters is evaluated by allocating different values for the percentage of vehicles, for which the quality of the data obtained from HTMC does not allow a correct identification. The simulation results recorded and compared after all scenarios are tested using the discrete simulation model.

The run parameters of the simulation model are presented in Table 1. Each simulation is replicated ten times. The results are the mean of the values from all replications. Before each replication, a warm period is used to avoid the transitory regime of the discrete simulation model.

**Table 1.** The simulation settings.

| Parameters | Value | Unit |
|---|---|---|
| Replication Length | 30 | Days |
| Number of Replications | 10 | - |
| Warm-up Period | 2 | Days |
| Hours per Day | 24 | Hours |
| Vehicles from EU with attribute values outside the legal limits and vehicles not from EU | 10 | % of total cargo vehicles flow |
| Number of places in Scaling process Queue | 6 | Cargo Trucks |
| Number of places in Second Verification Queue | 3 | Cargo Trucks |
| The Scaling process time (theoretical distribution) | $\lambda = 75$, $\sigma = 7$ Normal distribution | Seconds |
| The Second verification process time (theoretical distribution) | $\lambda = 50$, $\sigma = 3$ Normal distribution | Seconds |

A warm-up period was set for 2 days. The simulation results obtained in this period are not recorded and are not used in statistics. We consider that 2 days are enough to achieve a stationary regime for the queue system simulated.

The average values in the survey week were approximately 2400 cargo vehicles per day. Following historical data, some days may have higher or lower values. However, in the period before the winter holidays, values can be recorded and can reach an average value of approximately 2650 cargo vehicles per day. This value was used for a second baseline scenario.

In Table 2 are presented the main attributes of scenarios used to evaluate the influence of the uncertainty. This one is generated by the quality of data obtained from HTMC. In an ideal scenario, the uncertainty is zero and the smart equipment have information about vehicles in transit through the area monitored by it.

**Table 2.** The proposed attributes of simulation scenarios.

| No. of Scenario | Description |
|---|---|
| S10 | The base scenarios for initial configuration. Comparing the input data and length of the real queue on highway before BCP Nadlac the cargo trucks are generated according to Exponential (36 s) distribution |
| S11 | It is used for the proposed configuration with same distribution for simulation entities from S10 but with a rate of uncertainty of 10% (for approximately 10% of the cargo vehicles, the received data are incorrectly or the plate number is not found) |
| S12 | Same as in S11 but for a rate of uncertainty of 20% |
| S13 | Same as in S12 but for a rate of uncertainty of 30% |
| S14 | Same as in S13 but for a rate of uncertainty of 40% |
| S20 | Still following the base scenarios for initial configuration, the number of cargo trucks is increased with 10% in relation to the initial values analyzed (the entities are generated according to Exponential (32.7 s) distribution) |

**Table 2.** *Cont.*

| No. of Scenario | Description |
|---|---|
| S21 | It is used for the proposed configuration with the same distribution for simulation entities from S20 but with a rate of uncertainty of 10% |
| S22 | Same as in S21 but for a rate of uncertainty of 20% |
| S23 | Same as in S22 but for a rate of uncertainty of 30% |
| S24 | Same as in S23 but for a rate of uncertainty of 40% |

## 5. Results

The proposed simulation scenarios are used to evaluate the impact of the changing transit process at the exit point from a highway network placed before a real BCP located between Romania and Hungary. The values obtained for the base scenarios S10 are used to calibrate the model and to validate it by comparing the real queue length of cargo vehicles waiting before BCP and the simulation outputs. The results obtained are presented in Table 3.

**Table 3.** The simulation results.

| Parameters | S10 | S11 | S12 | S13 | S14 | S20 | S21 | S22 | S23 | S24 |
|---|---|---|---|---|---|---|---|---|---|---|
| Scaling process Queue length | 268 | 0.124 | 0.130 | 0.163 | 0.271 | 1054 | 0.223 | 0.202 | 0.276 | 0.426 |
| Second verification Queue length | - | 1.452 | 1.111 | 0.844 | 0.512 | - | 1.629 | 1.364 | 1.002 | 0.672 |
| Waiting time for Scaling process [hours] | 2.692 | 0.003 | 0.003 | 0.003 | 0.005 | 9.678 | 0.004 | 0.004 | 0.005 | 0.007 |
| Waiting time for Second verification [hours] | - | 0.021 | 0.018 | 0.014 | 0.010 | - | 0.024 | 0.020 | 0.016 | 0.013 |
| Usage rate of scaling teams [%] | 100 | 36 | 39 | 47 | 55 | 100 | 44 | 47 | 53 | 60 |
| Usage rate of second verification teams [%] | - | 92 | 85 | 78 | 67 | - | 94 | 90 | 83 | 71 |

The introduction of a new serving station in the modeled Queue System associated with the control area on exit point from Romanian Highway Network leads to a big improvement of serving parameters. The usage rate for scaling teams is reduced from 100% to an interval between 36 and 90%. The length of the queue is reduced for a real situation modeled in scenarios S10 and S20 from 268 and 1054 cargo vehicles to values under one.

The influence of uncertainty due to imported data leads to the variation of usage rates of a scaling team and second verification. According to access rules on the path to second verification using data received from HTMC presented in logical schemes from Figure 3, a cargo vehicle must meet a set of conditions. When these data are altered, and the level of uncertainty in the data increases, the number of vehicles directed to this new path decreases. Vehicles that are not found in the system database will be automatically directed to the weighing equipment. These variations are depicted in Figure 10.

The usage rates over 70% for operators from second verification teams are obtained in the case of small values for uncertainty. In these cases, the scaling equipment has the usage rate value under 50%. In these conditions, the maintenance operations also can be carried out without having a negative impact on vehicle traffic.

The increasing value of cargo vehicle flow with only 10% makes, in the actual situation, an important negative impact. The scaling process queue length is multiplied by 3.93 times and the waiting time for the scaling process is also multiplied 3.59 times.

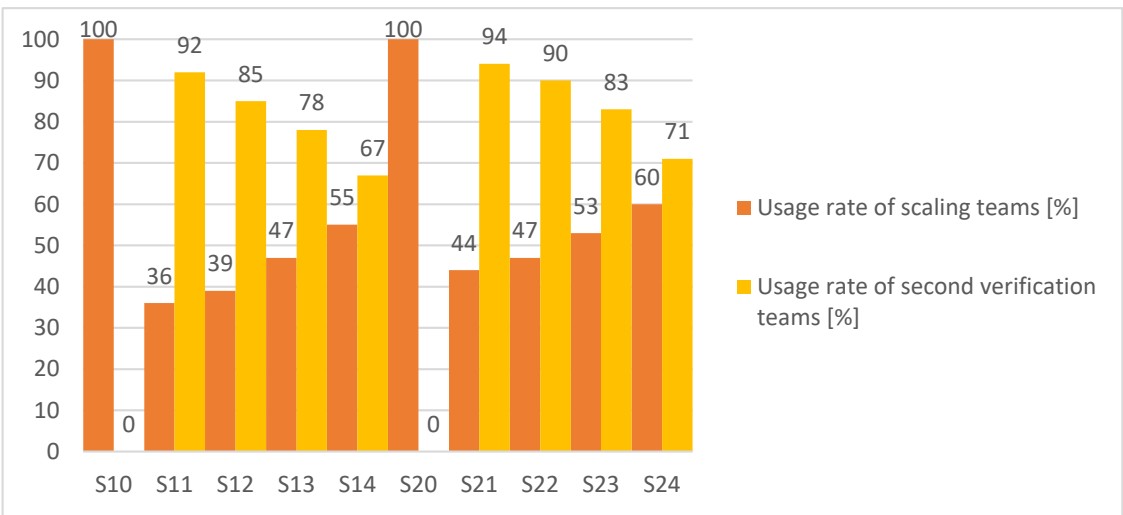

**Figure 10.** The usage rates of the serving teams.

## 6. Conclusions

The main objective of the paper was to evaluate the influence of uncertainty in a real situation. The authors are involved in the development process of new smart equipment for monitoring and controlling exit point activity from the Romanian Highway Network placed before a BCP. Internal regulation of the Highway Administration obliges a group of cargo vehicles to be scaled and verified in accordance with national rules. The lack of adequate equipment for the separation process of cargo vehicles, depending on the country of the owners and the technical characteristics, leads to the weighing process for all vehicles.

The authors propose an architecture for new smart equipment called Smart Hub and develop a logical model for a decisional process for vehicle separation. The input data will be obtained from highway sensors managed by a HTMC. The incorrect collection of vehicle data can have an unwanted effect on the way the system works. A simulation model was developed to evaluate the implication of the variation of data entropy. A set of scenarios was developed, and regarding the simulation results the following considerations can be expressed:

- For actual topology of the analyzed area, a small variation of cargo vehicle flow leads to large values for waiting time and a large number of vehicles in queue.
- The implementation of Smart Hub equipment has a positive impact with quantifiable effects.
- The uncertainty of data received from HTMC has a large impact over the usage rate of teams for scaling or for document control on a direct path to BCP. On the other hand, the impact on other attributes such as waiting time is limited.
- Results show the possibility of performing maintenance operations for scaling equipment (the usage rate of the two equipment is under the value of 50%) when the Smart Hub is implemented.

In conclusion, the objective of the research is accomplished by the authors. The development of a simulation model supposes a good understanding of the process realized in the analyzed area and actions to calibrate and validate the model using real data. The results can be compared with the results obtained for the case of police border control in Romanian BCPs [12]. Starting from the same number of serving stations (two border police teams versus two scaling equipment), the waiting time have similar values. The increasing of serving station leads to an important reduction of this parameter.

The model was developed for a specific situation for a Romanian BCP. It can be adapted for another BCP if we know the national rules and the topology of the BCP.

The results show the importance of data quality used as primary data in the separation process of cargo vehicles. It is certainly more important for the Highway Administration to prepare the maintenance operation of the scaling equipment and less for the highway user. The next steps involve the development of Smart Hub equipment and its implementation in the exit point from the highway, placed before BCP. The expected positive impact will help the authors and the Highway Administration to promote this solution to be implemented in all exit points placed before BCPs with other countries where the scaling process has an important negative impact.

**Author Contributions:** Project administration, S.B.; Resources, C.C.; Supervision, E.R.; Visualization, A.R.; Writing—original draft, F.R.; Writing—review & editing, A.R., C.O. All authors have read and agreed to the published version of the manuscript.

**Funding:** This research was funded by Romanian Ministry of Research, Innovation and Digitization contract number COD SMIS 2014+ 120419 and the APC was funded by the authors.

**Data Availability Statement:** The data presented in this study are presented in the paper. More information are available on request from the corresponding author. The primary data are not publicly available due to are part of a contract with the Authority that administers the highway network in Romania.

**Acknowledgments:** The research work contained in this paper was supported within the frame of the project Innovative platform for the provision and management of public services in smart cities—SMART HUB, Code SMIS 120419.

**Conflicts of Interest:** The authors declare no conflict of interest.

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
