# Peer review of "Evaluating the Influence of Data Entropy in the Use of a Smart Equipment for Traffic Management at Border Check Point"

_machines, doi:10.3390/machines10100937_

Round 1

Reviewer 1 Report

The article “Evaluating the influence of data entropy in the use of a smart equipment for traffic management at Border Check Point” is original, dedicated to the study of the influence of the uncertainty of the results of the identification of cargo vehicles on the parameters of the border checkpoint as a mass service system. The constructed logical model and the statistical methods testify to the scientific validity of the work. Research was carried out for a real situation considering real data. The results of the research were used in the process of implementing smart equipment at a real checkpoint, which confirms the practical value of this work.

             However, there are some inaccuracies in the text of the article. The article can be accepted for publication after correction in accordance with the following comments and recommendations:

1)       The work lacks the specifications of the sensors and cameras that were used to identify vehicles. It is appropriate to add this information. Why were these devices used? What is the data format at the output of the specified equipment?

2)       Some connections between blocks of the logical model (Figure 3) are built incorrectly. Thus, the logical block "Is a vehicle from UE with attributes values inside the legal limits" performs a check of a specific condition and must have two outputs: positive and negative results of checking the condition specified in the block. It is necessary to check the correctness of other blocks and the connections between them and correct the logical model.

3)       At the beginning of Section 4, it is stated that the flow of vehicles is modeled using a theoretical function. This article lacks an analytical expression for this function. It is also advisable to indicate the mathematical notation of the specified function with specific values of the arguments for at least one of the basic scenarios.

4)       It is not clear from the text of the article how a limited slot for a direct path to BCP is implemented in practice. This needs an explanation.

5)       Table 1 shows the "Warm-up Period" parameter. It is advisable to explain what exactly this parameter reflects. What distinguishes this period from other times of operation of the mass service system and how is its duration, 2 days - determined? What factors affect the duration of the "Warm-up Period"?

6)       In the last two lines of Table 1, in the cells of the values, it is necessary to first write the time without parentheses, indicating the units of measurement, and then indicate the type of distribution in parentheses. Thus, there will be consistency with the name of the corresponding parameters.

7)       It is not clear why the usage rate of second verification teams decreases when the uncertainty of the input data increases (Figure 8). It is necessary to comment on this trend, perhaps in Section 6.

8)       It is appropriate to add a comparative analysis of the results of this study and similar studies to the conclusions.

Reviewer 2 Report

The paper addressed interesting problem but the writing style needs to be properly revised to ensure clearer communications to readers. At this stage, it is difficult to assess the signifcance of the paper. 

The significance of the results depends heavily on the simulation design. More descriptions of simulation settings are needed. For example,
- In Table 1, how are the parameter setting refect real-world siutation? How have the parameters values been derived from real-world data collections?
- For the scaneiors in Table 2, did they cover all possibilities? Or did they represent events that frequently occurred? How did you increment from S10 to S20? Did the 10% increase occured in real-world scenarios? What were the justifications of using exponential distributions?

It's understanable that the paper is trying to solve real problems. However, the simulation model itself need to be explained on whether it is subjective to the situations in Romania or it can be generalized to other locations.

Reviewer 3 Report

Dear Authors,

Your paper on “Evaluating the influence of data entropy in the use of a smart equipment for traffic management at Border Check Point” describes development of an alternative traffic management system in order to ease and reduce time waiting time at the borders for vehicles that do not require scale verification. Despite high reasoning to conduct discrete simulation models for the support and analysis of the developed smart hub equipment, the presented results do not indicate support of that the logical model depicted in figure 3.

After careful verification I have come to the conclusion that the present manuscript does not suffice to the rigor needed in this type of publication.

My arguments are the following:

1. The present manuscript  lacks experimental detailing. What are the basis of the conducted simulations? The HTMC sensors are not described. The used scales are not described. The theoretical distribution mentioned step regarding empirical values are not introduced. The scenarios described in table 2 are not clearly described. The use of the Arena Simulation software is not presented. 

2. How does the calibration process take place in the simulation?

3. There is low reproducibility of the simulations.

4. Data entropy term used both in the title once in the text lacks detailing or context.

5. No data comparison has been made to the actual real flow for the same amount of time and included in the simulation. 

6. The uncertainty term extensively used is not clearly described, to sparse, with no actual data to indicate to support the uncertainty investigation.

Good luck in further development of your research.

Round 2

Reviewer 3 Report

Dear Authors,

I am glad that previous observations have contributed to the quality of the paper.

I will succinctly describe new information added to the manuscript:

- a detailed description for entropy (page 2)

- LPR component details (page 4)

- WIM component details (page 5)

- figure 5 (page 7)

- minimal description for software use (page 8)

- theoretical distribution for HTMC data (page 9 - 10)

- modelling method (page 11 - 12)

- calibration for the simulation (page 12)

- conclusion remarks  (page 14)

However, to bring addition readability, reproducibility to the paper I would recommend the following :

- the LPR need in depth description of the system that is responsible with data processing:

     - camera resolution, framerate in order to motivate the capability of processing the images for vehicle in motion with speeds from 0-300 km/h as mentioned in the manuscript

     - your system processing system (other systems in literature raspberry Pi/FPGA/Jetson Nano/others)

     - algorithm that is used to identify the license plates (examples in literature Caffe/ Yolo/MobileNet)

     - some description of the model used by you in your experiments

- WIM component should include a model of scale with description of resolution and precision of weighting process, the minimum and maximum range for weighting

      - some proper description where the weighting process may loose precision  or fail to return real data

- a more suitable description to figure 5 (a,b,c,d captions are ok)

- for reproducibility means a detailed description must be provided on how you have used the Arena Simulation software (for other researchers in the field, not the readers). If you do not want add such details, you should not mention this software in the next manuscript 

The quality of the presentation and readability of the present manuscript has improved since last submission. I highly recommend to take into account all suggestions above in order for the manuscript to achieve it's intended attention.   

I look forward to see the final manuscript. Good luck with your research!
